# Mekong delta much lower than previously assumed in sea-level rise impact assessments

P.S.J. Minderhoud [1,2], L. Coumou [1], G. Erkens[1,2], H. Middelkoop[1] & E. Stouthamer[1]

Deltas are low-relief landforms that are extremely vulnerable to sea-level rise. Impact assessments of relative sea-level rise in deltas primarily depend on elevation data accuracy and how well the vertical datum matches local sea level. Unfortunately, many major deltas are located in data-sparse regions, forcing researchers and policy makers to use low-resolution, global elevation data obtained from satellite platforms. Using a new, high-accuracy elevation model of the Vietnamese Mekong delta, we show that quality of global elevation data is insufficient and underscore the cruciality to convert to local tidal datum, which is often neglected. The novel elevation model shows that the Mekong delta has an extremely low mean elevation of ~0.8 m above sea level, dramatically lower than the earlier assumed ~2.6 m. Our results imply major uncertainties in sea-level rise impact assessments for the Mekong delta and deltas worldwide, with errors potentially larger than a century of sea-level rise.

[1] Faculty of Geosciences, Department of Physical Geography, Utrecht University, PO Box 80.1153508 TC Utrecht, The Netherlands. [2] Department of Subsurface and Groundwater Systems, Deltares Research Institute, PO Box 854673508 AL Utrecht, The Netherlands. Correspondence and requests for materials should be addressed to P.S.J.M. (email: P.S.J.Minderhoud@uu.nl)

Worldwide over 500 million people living in deltaic areas are increasingly exposed to flood hazards arising from climate extremes and relative sea-level rise[1,2]. The combined effect of relative sea-level rise (SLR), the result of absolute SLR and land subsidence, and reduced sediment aggradation on the delta surface causes many deltas to lose elevation relative to sea level[3,4]. Elevation loss increases the vulnerability to flooding and storm surges, and ultimately threatens deltas with permanent inundation. Considering the potentially large consequences, and to support policy makers in developing appropriate adaptation plans, impact assessments of relative sea-level rise are essential. The quality of such assessments primarily relies on the accuracy of available elevation data.

Digital elevation models (DEMs) have been subject of numerous studies on DEM accuracy, comparison between different DEMs[5,6], implications of DEM inaccuracy for river flood mapping[7,8] and required DEM correction for hydrodynamic modelling[9–11]. Few studies specifically focused on the effect of DEM accuracy on SLR impact assessments for low-lying flat coastal areas[12]. Global DEMs obtained from spaceborne platforms (e.g., Shuttle Radar Topography Mission (SRTM) DEM[13], TanDEM-X WorldDEM[14]) have a typical vertical accuracy of several meters. Moreover, their vertical resolution is too coarse to capture the subtle elevation differences, in the order of decimeters, required for accurate SLR impact assessments of flat delta regions[15,16]. Clearly, these global spaceborne DEM-products were not designed for such purpose, and the need for a better high-accuracy, open-access global DEM was recently voiced[17].

In contrast to spaceborne DEMs, airborne LiDAR measurements or geodetic surveys provide elevation data to create DEMs that do capture subtle, low-gradient elevation differences[7,8]. Unfortunately, many deltas around the world are located in data-sparse regions for which high-accuracy elevation data is not available, or—when existing—not publicly accessible. Consequently, global, open-access DEMs derived from spaceborne surveys are generally the only available elevation models, and, in spite of their low vertical accuracy and precision, many studies worldwide have used these to determine delta elevation above sea level[3], in their flood risk assessments or SLR and storm surge impact analyses[18,19].

Recently, the Multi-Error-Removed Improved-Terrain (MERIT) DEM was developed, in which multiple elevation errors existing in global DEMs have been removed in a consistent way[37]. The MERIT DEM was presented as especially helpful for flood inundation modeling, owing to its significantly more accurate elevation representation of flat regions. The MERIT DEM thus may improve the quality of future SLR assessments, although its vertical accuracy remains low when compared to magnitudes of SLR over the coming century.

A second issue arises from the fact that when global DEMs are used for assessments of relative elevation to local sea level, the elevations need to be converted from their reference to a global geoid to a local datum referenced to sea-level height[20]. Still, many global and regional studies neglect this crucial step, and implicitly assume zero elevation in the global geoid to represent local mean sea level. This is not correct, as the sea surface height is different from the geoid surface, due to e.g. water circulation and temperature-related variations in water density and resulting sea surface height. Also, different geoid models may locally represent different zero elevations. For example, vertical offsets up to five meters between the EGM96 (Earth Gravitational Model to which the SRTM DEM is referenced) and the newer EGM08 geoid model have been documented in Turkey[22], illustrating the potential magnitude of vertical error. Wrongly considering the geoid datum of global DEMs to represent local sea level, or not converting DEM elevation to local zero (tidal) datum, potentially introduces large errors in elevation above sea level, and hence in impact assessments.

The Vietnamese Mekong delta has one of the largest and seemingly lowest elevated delta plains in the world[3]. While ongoing land subsidence increases the rate of relative SLR[23–25], the sediment load of the Mekong river to counterbalance relative (SLR) with sediment accretion on the delta plain is dwindling due to upstream dam construction[26,27] and decreased hurricane activity in the Mekong catchment[28]. In spite of the awareness of the low vertical accuracy and resolution of global DEMs within a large scientific community[15,16], several studies did use SRTM elevation data[29–31] to assess potential impacts of RSLR for the Mekong delta (Fig. 1). None of these studies performed any conversion or correction of vertical datum, nor mentioned the implications of omitting this. Resulting products from such studies have been adopted in policy advisory reports[32–36](Fig. 1b, c).

For our study we have acquired a large dataset of elevation points from a detailed topographical map of the Vietnamese Mekong delta referenced to Vietnamese geodetic datum (with mean sea level measured at the Hon Dau tide gauge as zero datum); these elevation data were previously unavailable for users outside Vietnamese government institutes. We created a new elevation model for the delta based on these elevation points (Topo DEM), and compared it to the SRTM and the MERIT DEMs, both vertically referenced to the EGM96. A comparison between Vietnam's Hon Dau datum and the newer EGM08 geoid revealed a mean elevation bias of +0.890 m[21], suggesting similar or potentially vertical larger offsets with the EGM96. We therefore performed three independent analyses to evaluate the elevation accuracy of all three DEMs. To evaluate absolute elevation, we used an independent dataset of elevation benchmarks referenced to Hon Dau datum. As additional check of DEM consistency, we analyzed the relative elevation patterns by comparing the DEMs to a geomorphological map and a dataset of tide-dominant flood occurrences. Finally, we assessed the implications of using different DEMs and neglection of vertical datum conversion for relative SLR impact assessments. We demonstrate that the Mekong delta has a much lower elevation above local sea level than previously assumed. Our findings underscore the large uncertainties associated with the use of spaceborne DEMs and stress the need of correct conversion of such DEMS to local tidal datum. Moreover, our results imply potential, similarly large errors in SLR impact assessments for other data-sparse deltas and coastal plains on Earth.

## Results

**Elevation models**. The new Topo DEM (Fig. 2, Supplementary Fig. 2 for color-blind-friendly version version) shows the topography of the Mekong delta, with higher areas in the NW, upstream part of the delta, along the Mekong river branches and at the coastline in the SE. Low areas occur toward the west coast and at distal locations in the delta plain away from the river system and the SE coastline. A large area in the SW part of the Mekong delta lies only several decimeters above MSL.

The spaceborne SRTM and MERIT DEMs (Fig. 3, Supplementary Fig. 3 for color-blind-friendly version) show remarkable differences in elevation of the Mekong delta when compared to the Topo DEM. The average elevation of the delta plain above vertical datum (excluding areas with bedrock outcrops) is 2.6 m in the SRTM DEM and 3.3 m in the MERIT DEM, while it is 0.82 m in the Topo DEM. A clearly visible NE-SW oriented striping pattern dominates the SRTM DEM; in MERIT DEM this noisy striping has been largely removed, but some remains of major banding in elevation are still present. The Topo DEM does not show these features.

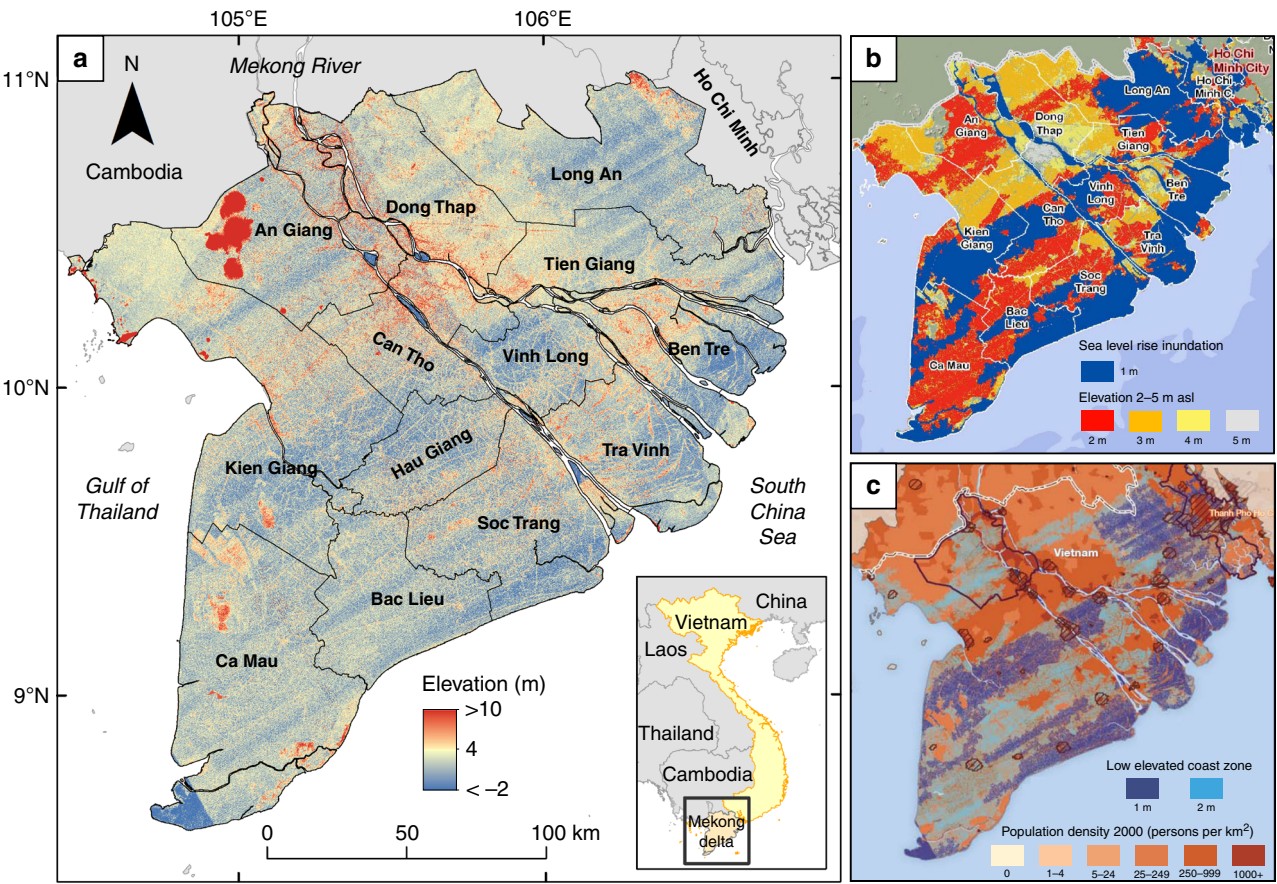

**Fig. 1** Spaceborne elevation of the Mekong delta and derived inundation maps. **a** Shuttle Radar Topography Mission (SRTM) Digital Elevation Model (DEM) of the Mekong delta in Vietnam. **b**, **c** Inundation maps following sea-level rise based on the SRTM DEM containing effects of striping and other height errors. **b** Inundation zones with meters sea-level rise[29], copyright by 2008 ICEM. Used by written permission. This map was included in several policy documents, amongst others, the Asian Development Bank[36], Mekong River commission technical paper[32]; World Wildlife fund risk assessment Mekong[34] and the Dutch-Vietnamese Mekong Delta Plan[35], which is now a leading document for major Worldbank projects in the delta. **c** Low elevated coast zones with population density[30,38], copyright by 2008 CARE International and UN University. Used by permission. This map was included in, amongst others, the risk assessment Mekong by the World Wildlife Fund[34]

The high horizontal resolution (30 × 30 m) SRTM DEM, and to a lesser extend also the MERIT DEM (94 × 94 m), enables to capture small-scale topographical features, such as natural levees and beach ridge remnants, elevated roads and cities. These details are not present in the Topo DEM as its spatial resolution is too coarse (500 × 500 m).

Two elevation profiles through the delta show the typical differences between the DEMs. One cross-section runs parallel to the main Mekong river branches spanning from the upstream river apex in the NW to the coastline in the SE (Fig. 3, profile A–A′) and the other cross-section is NE-SW orientated, perpendicular to the river branches (Fig. 3, profile B–B′). The elevation profiles of the SRTM DEM show an erratic topography with excursions high above the average delta plain elevation to several meters below sea level. The elevation profiles of the MERIT DEM show a more smoothed pattern with fewer excursions in both directions. Especially the lower end excursions present in the SRTM DEM are no longer present, resulting in higher mean elevations in the MERIT DEM. Compared to both spaceborne DEMS, the elevation profiles of the Topo DEM are smoother and much less erratic and show a, generally, much lower elevation of the delta surface.

The elevation statistics at provincial level are given in Table 1. In both the Topo DEM and the SRTM DEM, the province Hau Giang has the lowest land surface, closely followed by Kien Giang,

Bac Lieu, and Ca Mau for the Topo DEM and Bac Lieu, Soc trang and Vinh Long for the SRTM DEM. In the MERIT DEM Ca Mau is lowest elevated, followed by Vinh Long and Tra Vinh and Bac Lieu. The upstream-located provinces are generally higher in all DEMs. Average elevation of An Giang and Dong Thap is highest in the Topo DEM; Dong Thap is also highest in both the SRTM and MERIT DEM.

**Absolute elevation validation using benchmarks.** The absolute elevations of 69 national benchmarks (referenced to Vietnam's Hon Dau geodetic datum) located across the entire delta were compared to the corresponding elevation given by the DEMs (Fig. 4; Supplementary Information (SI), Supplementary Table 2; Supplementary Fig. 4). In the SRTM DEM the mean elevation at the benchmark locations deviates by +2.0 m with a standard deviation (SD) of 2.9 from the mean elevation of the benchmarks. Compared to the SRTM DEM, the residuals of the MERIT DEM show a larger mean deviation of +3.0 m with a lower SD of 1.3 m. The large mean deviations indicate a structural overestimation of the geodetic surface elevation in the SRTM and MERIT DEMs, which is partly the result of the difference between Hon Dau datum and the EGM96 to which both DEMs are referenced. The residuals of the Topo DEM have a much smaller mean deviation of +0.2 m (SD = 0.7 m), and they resemble a normal distribution

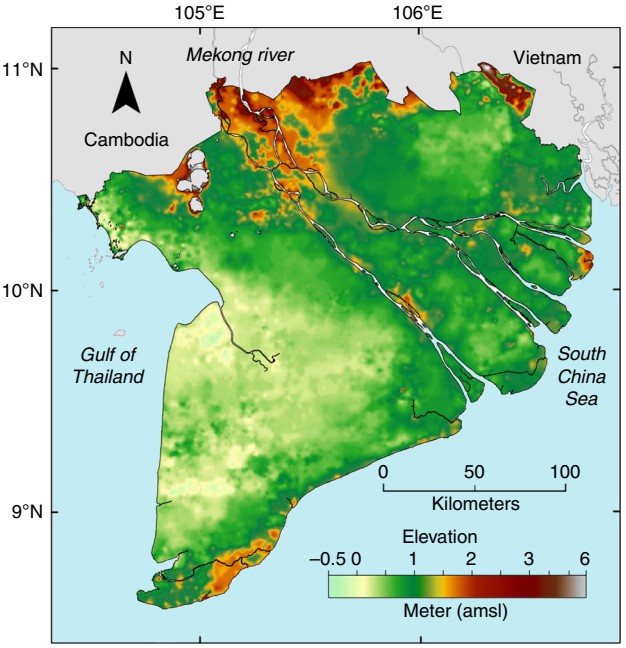

**Fig. 2** New elevation model of the Vietnamese Mekong delta. The digital elevation model (Topo DEM[59]) is interpolated based on nearly 20,000 topographical elevation points (Supplementary Fig. 1) and gridded at 500 m × 500 m. Supplementary Fig. 2 is a color-blind-friendly version of this figure

with 57% confidence (Anderson-Darling normality test). The offset of 0.2 m may well be the result of the elevation difference (~−0.3 m, Supplementary Fig. 5) between the benchmarks and actual surface elevation as presented by the Topo DEM, but also falls within the interpolation error (SD = 0.16 m) of the Topo DEM. The individual deviations in elevation between the benchmarks and the Topo DEM are in the order of decimeters up to 2 m (mean absolute deviation is 0.6 m) (Fig. 4). The residuals do not show a specific spatial pattern over the delta and, therefore, may stem from comparing elevation of point locations to average grid cell elevation (500 × 500 m). For the SRTM and the MERIT DEM, the individual absolute deviations are much larger (mean absolute deviation of 2.6 and 3.0 m). Whereas the SRTM and the MERIT DEM seem to systematically overestimate actual delta surface elevation, partly as a result of a different vertical datum, the Topo DEM appears to represent the geodetic elevation of the delta surface at decimeter accuracy.

**Relative elevation validation using geomorphology.** Geomorphological units are, by nature, characterized by their typical relative elevation compared to adjacent geomorphological units, as a result of their specific depositional environment (Supplementary Fig. 6). To use this principle as test for the consistency of the DEMS, we adopted the geomorphological map of the Mekong delta[39] (Supplementary Fig. 7) as a proxy for relative elevation. Table 2 shows the mean elevation of each geomorphological unit on this map according to each DEM, and—in qualitative terms— its expected elevation relative to other units within the same category, following from its geomorphological interpretation. All DEMs show the expected relative elevation differences between the three main categories, with the older Pleistocene deposits having the highest mean elevation, followed by the Holocene upper alluvial delta plain and, subsequently, the lower coastal delta plain with the lowest average elevation.

The Topo DEM represents the expected relative elevation of different geomorphological units generally well within each of the main classes. Within the upper alluvial delta plain, the natural levees and channel bars are the highest elevated units, while the flood basin and the swamps are correctly indicated by the Topo DEM as the lowest areas. Channel bars are elevated as high as natural levees and this is probably because they actually represent large vegetated islands within river branches, which feature a similar depositional environment as natural levees. In the lower coastal delta plain, inland marshes are the lowest unit, with an average elevation of 0.34 m above sea level. The coastal plain elevation is around half a meter above datum, whereas sand spits are, as expected, the highest geomorphological unit in the coastal zone. The elevation of "relict beach ridge or sand dune" on the Topo Dem is remarkably lower than expected. This is likely due to the low spatial resolution of the Topo DEM (500 m × 500 m) that does not capture such small geomorphological features. Moreover comparison with aerial photographs revealed that the geomorphological map does not always match with the exact location of these beach ridges, and has a tendency to overestimate their spatial extent.

The SRTM and the MERIT DEM rightly reflect the highest geomorphological units within each main class, which are weathered land, natural levees and sand spits. However, the expected order of relative elevation ranking among the lower elevated geomorphological units is absent, and all classes have a similar mean elevation. Especially for the lower delta plain, the SRTM DEM hardly shows any elevation differentiation between the geomorphological units. In the MERIT DEM the expected order in the lower delta plain is even reversed, with the geomorphological units expected to have the lowest relative elevation, having higher mean elevation (see Supplementary Fig. 8 and Supplementary Table. 3–5 for elevation statistics for each geomorphological unit in each DEM).

**Relative elevation validation using tide-dominated floods.** The second method used to validate relative elevation was based on the assumption that tide-dominated floods occur more often in lower areas. Tidal flooding is the dominant factor controlling the number of inundations and therefore, when an elevation model is correct, increasing flood occurrence is expected to correlate with decreasing elevation. Therefore for each DEM the mean elevation of the delta surface was determined for all flood occurrences during the period 2007–2011[15] (Supplementary Fig. 9) in the coastal zone of the southwest part of the Mekong delta (Supplementary Fig. 10). Total inundated areas and boxplots showing the distribution of elevation points per flood occurrence are given in the SI (Supplementary Fig. 11). The elevations of both the MERIT and the Topo DEM show the expected correlation of decreasing elevation with increasing flood occurrence with a continuous trend, while the elevations in the SRTM DEM only weakly show this correlation (Supplementary Fig. 11).

In addition to a relative elevation validation, this analysis also contributes to the absolute elevation validation of the DEMs. Tide-dominated floods in the delta occur when the average high tide level is exceeded as a result of spring tides and storm tides. Most of the coastal flooding occurs in the southwest of the Mekong delta plain toward the Gulf of Thailand, where the tidal range is 40 cm[40]. The absolute elevations of the Topo DEM of areas that experience regular tide-dominated floods are within decimeters of the average tidal range and MSL, which is realistic. The MERIT and SRTM DEM, referenced to EGM96 rather than a tidal datum, provide absolute elevation values that are at least 1 m higher, and such elevations are unlikely to experience regular coastal flooding considering the small tidal range.

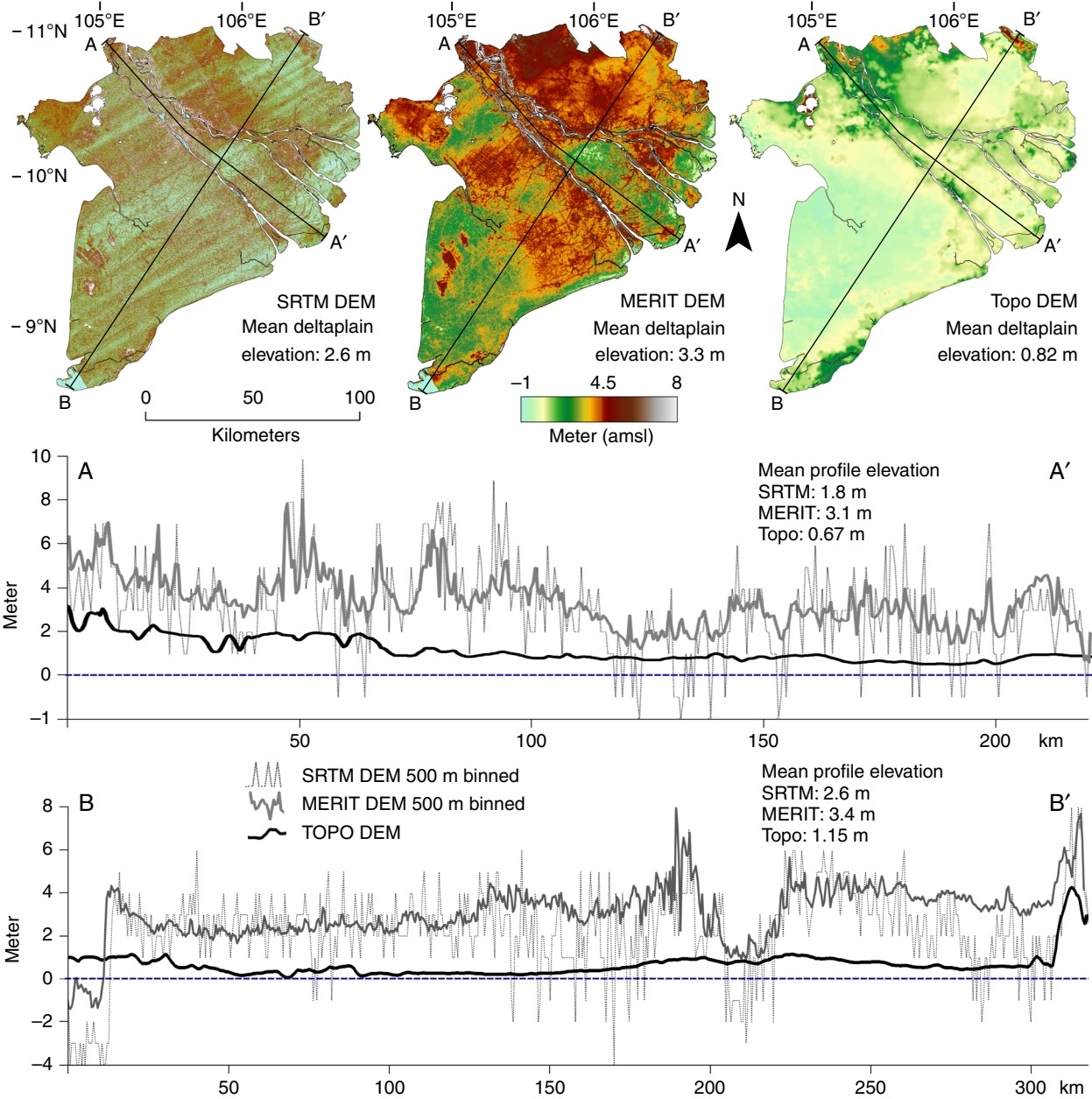

**Fig. 3** Digital elevation models of the Mekong delta with elevation profiles. The elevation profiles of the SRTM and MERIT Digital Elevation Models (DEMs) was binned (mean elevation) in 500-m segments to match the spatial resolution of the Topo DEM to allow visual comparison. Mean elevation of the DEMs was calculated for the delta plain only, excluding the higher elevated bedrock outcrops in the northwest of the delta. Supplementary Fig. 3 is a color-blind-friendly version of this figure

**Area below sea level following relative sea-level rise**. The high vertical precision of the novel Topo DEM and the transposed MERIT DEM (original MERIT DEM minus 2.5 m, see methods) allowed for impact analyses of relative SLR (hereafter: SLR) at decimeter scale. The area below sea level for SLR scenarios of 20, 50, and 80 cm above mean sea level (MSL) based on the Topo and the transposed MERIT DEM is shown in Fig. 5. Based on the Topo DEM, 20 cm SLR results in 6% of the total delta plain area below sea level. This area progressively increases to 29 and 54% of the total delta plain after a SLR of 50 and 80 cm, respectively. Based on the transposed MERIT DEM, 32, 41, and 51% of the total delta plain area falls below sea level under 20, 50, and 80 cm SLR.

The estimated area per province below sea level following relative SLR based on the Topo DEM is shown in Fig. 6a (See

Supplementary Table 1 for elevation statistics per province). Based on these results, the provinces can be divided into two main groups. The first group comprises the four provinces most prone to SLR: Bac Lieu, Ca Mau, Hau Giang, Kien Giang, which are all located in the southwestern part of the Mekong delta. Where a SLR of 20 cm already moderately affects the provinces of Ca Mau and Kien Giang, the largest impact occurs under a SLR of 20–60 cm. Half a meter of SLR will result in over 50% of the delta area of each of these provinces to fall below sea level. The second group consists of provinces located in the middle and north-eastern part of the delta, which become mostly affected when SLR reaches 60–90 cm. The province of Soc Trang is an exception that falls between both groups. This province is most affected by a SLR of 30–90 cm. The estimated number of people below sea-level following SLR show a similar pattern (Fig. 6b). With a SLR of

**Table 1 Mean elevation of provinces and the whole Vietnamese Mekong delta**

| Province | Topo DEM | SRTM DEM | MERIT DEM |
|---|---|---|---|
| An Giang | **1.42** | 3.3 | 3.8 |
| Đồng Tháp | 1.41 | **3.7** | **4.4** |
| Long An | 1.07 | 2.4 | 3.9 |
| Bến Tre | 0.95 | 2.6 | 3.1 |
| Vĩnh Long | 0.94 | 2.1 | 2.6 |
| Tiền Giang | 0.85 | 2.8 | 3.6 |
| Trà Vinh | 0.79 | 2.5 | 2.8 |
| Cần Thơ | 0.72 | 2.9 | 3.9 |
| Sóc Trăng | 0.68 | 2.0 | 3.5 |
| Cà Mau | 0.59 | 2.3 | **2.4** |
| Bạc Liêu | 0.50 | 2.0 | 2.9 |
| Kiên Giang | 0.39 | 2.5 | 3.1 |
| Hậu Giang | **0.38** | **1.9** | 3.4 |
| Entire Mekong delta | 0.82 | 2.6 | **3.3** |

Provinces are ranked from **highest** to *lowest* elevation according to the Topo Digital Elevation Model (DEM). The mean values are given for the Topo, SRTM and MERIT DEMs with one additional decimal than present in the input data to distinguish mean elevation differences. Bedrock outcrops, rivers and islands were excluded from this analysis. Province boundaries are shown in Fig. 1A. Additional elevation statistics are given in Supplementary Table 1

20 cm, an area inhabited by ~1 million people (6% of the total population) falls below sea level (mainly located in the first group of provinces). With a SLR of 50 cm this area increases to an area inhabited by approximately ~4.7 million people (27% of the total population).

When using the SRTM DEM, a 1-m SLR causes 31% of the Mekong delta plain to fall below sea level. In the MERIT DEM, the transposed MERIT DEM and the Topo DEM, this is area is, respectively, 2, 59, and 75% (Fig. 7). This corresponds to an estimated number of affected people of 5.1 million (29% of total population) based on the SRTM DEM, 0.3 and 9.7 million (2 and 55% of the total population) based on the MERIT and transposed MERIT DEM, and 12.3 million (70% of the total population) based on the Topo DEM (Table 3). Spatially, the affected area also largely differs between the SRTM, MERIT, and the Topo DEM.

## Discussion

In each elevation validation analysis, the Topo DEM performed superior to both spaceborne DEMs. The SRTM and the MERIT DEM are both vertically referenced to the global EGM96 geoid, while the Topo DEM and the national benchmarks, used to evaluate absolute vertical elevation, are referenced to Hon Dau datum. Partly for this reason, the SRTM and MERIT DEM

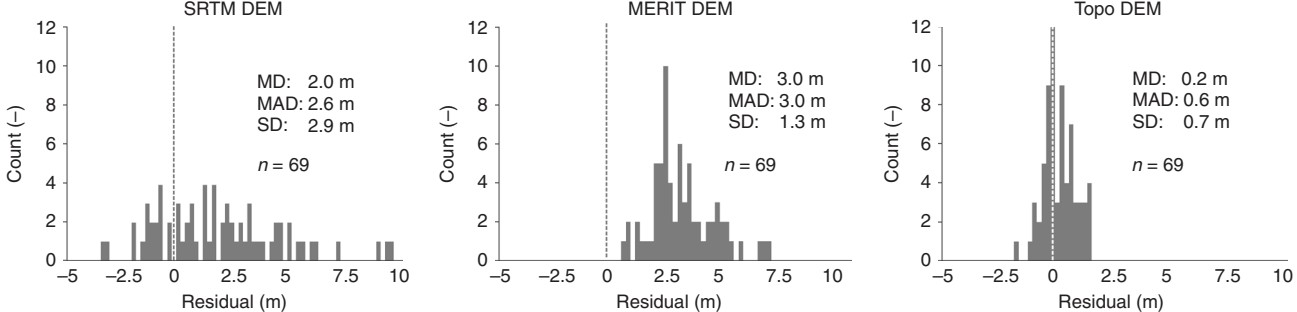

**Fig. 4** Elevation offsets between elevation models and national benchmarks. Histograms of elevation differences between the SRTM, MERIT, and the Topo Digital Elevation Models (DEMs) and national benchmarks in the Mekong delta. A positive residual means DEM elevation exceeds benchmark elevation. MD = mean deviation (difference in means), MAD = mean absolute deviation (mean of individual differences), SD = standard deviation of residuals

**Table 2 Elevation comparison of geomorphological units in the Mekong delta**

| Category and age | Geomorphological unit | Expected relative elevation | Mean elevation (m) | | |
|---|---|---|---|---|---|
| | | | Topo DEM | SRTM DEM | MERIT DEM |
| Pleistocene (and older) deposits | Weathered land | Very high | **3.27** | **6.4** | **6.9** |
| | Undifferentiated deposits | Higher | 1.84 | 3.1 | 4.7 |
| | Alluvial apron | Higher | **1.38** | **2.8** | **4.2** |
| Upper delta plain Holocene alluvial deposits | Flood basin | *Benchmark* | 0.85 | 2.8 | 3.5 |
| | Bank: natural levees and crevasse splay | Highest (++) | 1.38 | **4.1** | 4.7 |
| | Channel bar | Higher (+) | 1.41 | 3.2 | 5.1 |
| | Abandoned channel | Higher (+) | 0.91 | 2.7 | **3.5** |
| | Swamp | Lower (−) | 0.85 | 2.6 | 3.7 |
| | Back swamp | Lower (−) | **0.66** | **2.4** | **3.5** |
| Lower delta plain Holocene coastal deposits | Tidal flat | *Benchmark* | 0.99 | 2.2 | 1.5 |
| | Sand spit | Higher (+) | **1.14** | **2.7** | 2.8 |
| | Relict beach ridge or sand dune | Higher (+) | 0.84 | 2.2 | **1.6** |
| | Mangrove marsh | Equal (±) | 0.97 | 2.4 | 2.3 |
| | Salt marsh | Equal (±) | 0.83 | **2.1** | **3.1** |
| | Coastal plain | Lower (−) | 0.53 | **2.1** | 2.9 |
| | Marsh (inland) | Lowest (− −) | **0.34** | **2.7** | 2.7 |

The expected relative elevation of the Pleistocene and Holocene deposits are compared to the elevation according to the Topo, SRTM, and MERIT DEMs; within the Holocene deposits relative elevation is given compared to the elevation of the *benchmark (italic underlined)* geomorphological unit, respectively, *flood basin* for the upper delta plain and *tidal flat* for the lower delta plain. Expected relative elevation was based on typical elevation characteristics of the depositional environment corresponding to the geomorphological unit (see Supplementary Fig. 6). **Highest** (bold underlined) and *lowest* (bold italic) mean elevations in each category are highlighted. The mean values are given with one additional decimal than present in the input data to distinguish mean elevation differences. See Supplementary Fig. 7 for the geomorphological map of the Mekong delta[39] and Supplementary Tables 3–5 for additional statistics

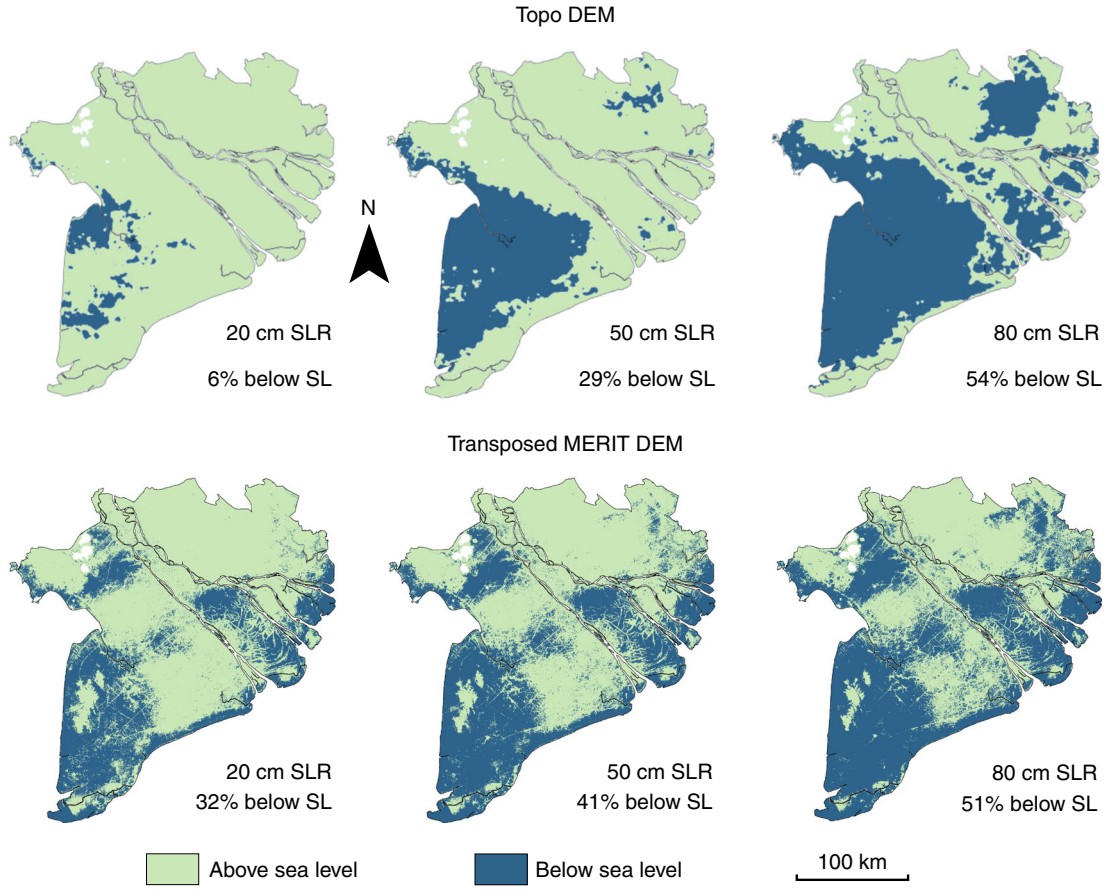

**Fig. 5** Projections of area below sea with relative sea-level rise scenarios. Area below sea level (SL) for, respectively, 20, 50, and 80 cm relative sea-level rise (SLR) based on the Topo Digital Elevation Model (DEM) and the transposed MERIT DEM. The transposed MERIT DEM matches the mean delta elevation of the Topo DEM and was created by subtracting 2.5 m from the original MERIT DEM. The percentages give the area of the delta plain below sea level (excluding bed rock outcrops)

overestimate the absolute surface elevation at the national benchmarks on average by 2 and 3 m, while the Topo DEM shows almost no absolute elevation overestimation (0.2 m). This example clearly illustrates the large difference that can exist between global geoid models and local geodetic datum and reveals the potential major errors that may arise in assessments of SLR impacts when vertical datum of global spaceborne DEMs is not converted to a local tidal datum. Clearly such assessments should at least include local elevation data to correct potential vertical biases of such DEMS for specific coastal regions or deltas, as was demonstrated by transposing the MERIT DEM.

In terms of relative elevation, the Topo DEM shows the expected relative elevation differences between geomorphological units. Both SRTM and MERIT DEMs fail to represent the expected elevation differences based on terrain geomorphology. In contrast to the SRTM DEM, both the MERIT and the Topo DEM correlate well with tide-dominated flood occurrence. The high horizontal resolution of the SRTM and the MERIT DEMs does allow distinguishing small, individual topographical features such as dikes, roads and natural levees, which are not present in the Topo DEM. However, at delta scale, the Topo DEM seems to represent the actual delta plain elevation relative to local sea level far better than the SRTM and the MERIT DEM, both in terms of absolute elevation and relative elevation.

We used the absolute average elevation difference between the MERIT DEM and the Topo DEM to transpose the MERIT DEM in an attempt to remove the vertical bias, potentially stemming from differences in vertical datum and to mimic a conversion to local tidal datum. Although this improved the applicability of the MERIT DEM, still the analysis based on the transposed MERIT DEM resulted in an underestimation of the vulnerability of the Mekong delta to SLR compared to the analysis using the Topo DEM. These results confirm the observation that, even when correctly converted to local tidal datum, SRTM-based estimates have the tendency to underestimate SLR impacts for lower elevated coastal areas due to DEM errors[12].

Analysis of the Topo DEM shows that apparently the Mekong delta plain is elevated much lower above local sea level than previously concluded based on SRTM elevation data[3,29,31,41]. According to the Topo DEM, the Mekong delta plain has a mean elevation of 0.82 m above Hon Dau tidal datum, which is only one third of the SRTM DEM's mean elevation of 2.6 m, and even less compared to the MERIT DEM's mean elevation of 3.3 m. Both SRTM and MERIT DEMs are referenced to EGM96 datum, of which zero elevation was in previous studies assumed to match sea level, while actual local sea level seems to be much higher. This revelation means that the delta is even more vulnerable to SLR than previously foreseen, and moderate estimates of absolute SLR (~40 cm by 2100[42]) may already result in 25% of the delta falling below sea level by the end of the century. Moreover, as the delta itself is subsiding at increasing rates following groundwater overexploitation[24], with present delta-average rates exceeding 1.1 cm per year, large parts of the delta may face submersion already during the coming decades.

Elevation above sea level is a key factor to assess the vulnerability of deltas to risks arising from SLR and land subsidence,

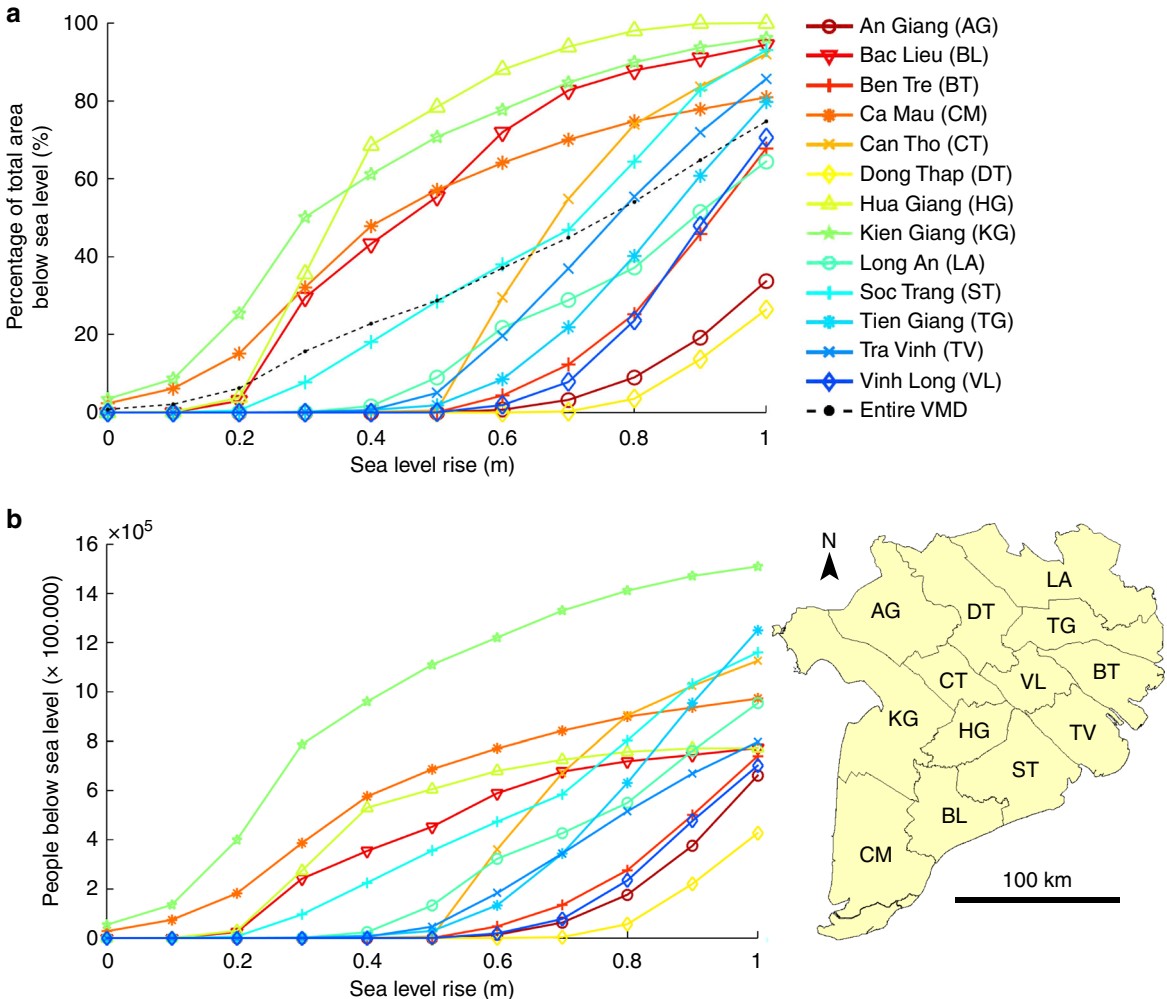

**Fig. 6** Area and estimated number of inhabitants affected by sea-level rise. Delta area (**a**) and estimated number of inhabitants (**b**) below sea level following sea-level rise up to 1 meter based on the Topo digital elevation model

which both decrease the relative elevation of a delta. The superior vertical accuracy of the Topo DEM over the SRTM DEM enables more reliable SLR impact assessments for the Mekong delta. The cumulative area that falls below sea level following a SLR of 1 m is a factor two larger according to the Topo DEM compared to the SRTM DEM, respectively, 75% versus 31% of the total delta plain (Table 3). Also the spatial patterns of the affected area are completely different between the two elevation models (Fig. 6). In spite of the improvements of the MERIT DEM when compared to the STRM[37], direct (and incorrect) use of the MERIT DEM as a measure of elevation above sea level would wrongly suggest that 1 m SLR only inundates 2% of the total Mekong delta plain, as the mean elevation of MERIT DEM is even higher than the SRTM DEM. The vertically transposed MERIT DEM by −2.5 m to match the average elevation of the Topo DEM still results in large differences in terms of delta area inundated and number of people impacted, when compared to using the Topo DEM. According to the Topo DEM over 12 million people (>70% of the total population of the delta) live in areas which will fall below sea level following a SLR of 1 meter, which doubles the number of earlier analyses using the SRTM DEM[29] (~5 million people, 29% of the total). The above numbers reveal the large differences between SLR impact assessments of a single delta and their strong dependency on the quality, accuracy and vertical datum of the used elevation model.

With rising sea level, present and future land surface elevation of a delta above sea level in a way determines time remaining before delta drowning. The apparent much lower elevation of the Mekong delta above local sea level means that there is much less time to implement mitigation or adaptation strategies than previously realized by the international research community. The southwestern part of the delta with the provinces of Bac Lieu, Ca Mau, Kien Giang, and Hua Giang is the lowest and therefore most vulnerable part of the delta. However, it is hard to predict when and where exactly in the delta the surface elevation will fall below sea level as this does not depend solely on present elevation, but also on the combined effect of climate change-driven SLR and subsidence. Furthermore, sedimentation of clastic and organic sediments at the delta surface, in turn, can increase elevation of the delta plain and must also be considered. The sum of these factors determines when a certain part of the delta plain will lose its elevation above sea level. Present rates of local eustatic SLR in the Mekong are ~3.3 mm per year[43]. Land subsidence in the Mekong delta is the result of the cumulative effect of various subsidence drivers[44]. Groundwater extraction causes subsidence rates to exceed 25 mm per year in certain areas[24], while natural compaction of young Holocene sediments contributes up to 20 mm per year to subsidence rates in the coastal zone[25]. Oppositely, sedimentation in coastal mangrove forests with sufficient sediment supply amounts to rates of ~36 to ~67 mm per year[45], while

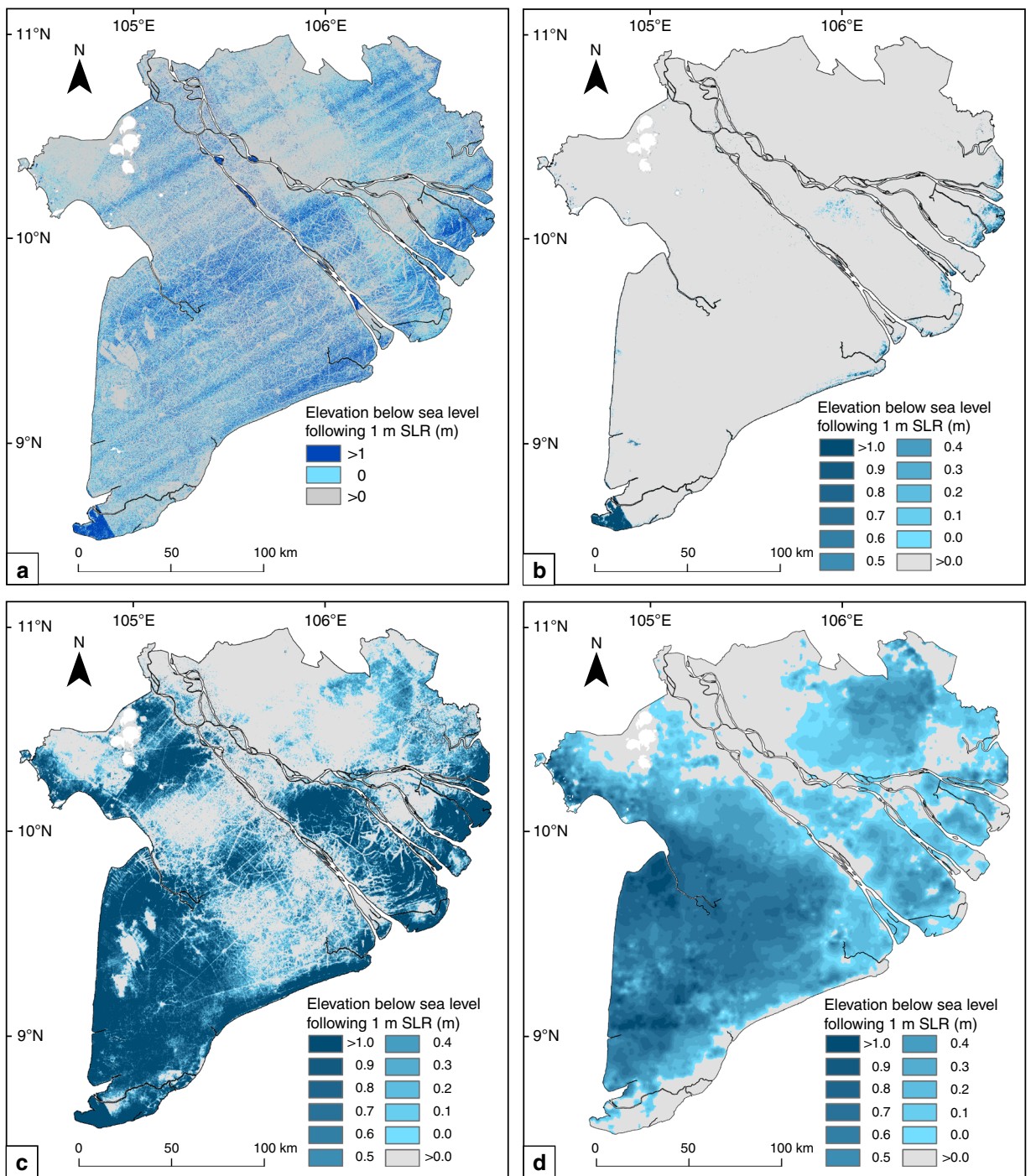

**Fig. 7** Projections of area below sea level following sea-level rise. Projections of relative sea-level rise (SLR) are based on the **a** SRTM DEM, **b** the MERIT DEM, **c** the transposed MERIT DEM, and the **d** Topo DEM. The transposed MERIT DEM matches the mean delta elevation of the Topo DEM and was created by subtracting 2.5 m from the original MERIT DEM. Total area of the delta plain below sea level with 1 m SLR is 31% for the SRTM DEM, 2% for the MERIT DEM, 59% for the transposed MERIT DEM, and 75% for the Topo DEM

estimates of average sedimentation rate on the Mekong flood-plains amount ~6 mm per year[146]. Both subsidence and sedimentation can spatially be highly variable and, as not all processes are yet mapped for the entire delta, a more detailed assessment of the timing of complete elevation loss in the delta as a result of SLR is currently not possible. Nevertheless, given the above numbers, it is likely that within the coming generations large parts of the Mekong delta will fall below sea level and even larger parts will experience increased nuisance flooding well before that time[47]. Whether or not this also means permanent inundation

also depends on the level of flood protection and ability to manage surface water and sedimentation through for example polder systems, like in the Dutch Rhine-Meuse delta. However, given its large extent, it will be an impossible task to protect the entire Mekong delta plain against drowning, and thus difficult delta management choices to prioritize flood protection will arise.

This study clearly shows the importance of accurate elevation data, such as DEMs derived from detailed topographical elevation data or LIDAR campaigns, and correctly referencing to local geodetic datum and local sea level for SLR assessments in low-

**Table 3 Area and estimated people below sea level with 1 m sea-level rise**

| | SRTM DEM | | MERIT DEM | | Transposed MERIT DEM | | Topo DEM | |
|---|---|---|---|---|---|---|---|---|
| | 0 m SLR | 1 m SLR | 0 m SLR | 1 m SLR | 0 m SLR | 1 m SLR | 0 m SLR | 1 m SLR |
| Delta plain area below SL (×1.000 km$^2$) | 5.5 | 11.9 | 0.2 | 0.8 | 10.2 | 23.9 | 0.3 | 28.5 |
| Delta plain area below SL (% of total area) | 14% | 31% | 1% | 2% | 25% | 59% | 1% | 75% |
| Estimated number of people below SL (×10$^6$) | 2.4 | 5.1 | 0.1 | 0.3 | 4.1 | 9.7 | 0.1 | 12.3 |
| Estimated number of people below SL (% of total population) | 14% | 29% | 1% | 2% | 23% | 55% | 1% | 70% |

Delta plain area and estimated number of people below sea level (SL) for 0 and 1 m relative sea-level rise (SLR) according to different DEMs for the Vietnamese Mekong delta. Total delta plain area is based on the area of the flat, deltaic surface excluding high bedrock outcrops. All DEMs contain elevation values below zero, which is used to estimate area and people below sea level with 0 m SLR as reference

lying deltas and coastal plains worldwide. Such assessments are crucial to the development of sustainable delta management strategies. In Vietnam, the government is aware of the delta's low elevation; Vietnamese SLR assessments following climate change scenarios in internal governmental reports show inundation patterns and magnitudes similar to our analyses using the Topo DEM[48]. Unfortunately, neither such internal documents nor the important underlying elevation data have sufficiently found their way into the global climate adaptation studies.

Presently available spaceborne DEMs suffer from accuracy problems in the order of meters, and will lead to major errors in assessments of coastal drowning[12]. Remarkably, the MERIT DEM visually shows a less noisy and banded elevation when compared to the SRTM DEM, but yields no improved estimate of the Mekong delta relative elevation. Whenever available, elevation data from local geodetic surveys, and the relation between the used reference datum and local sea level should be obtained, for direct use, or to vertically adjust global DEMs and relate it to local sea level as a first-order estimate.

The acquisition of accurate elevation datasets based on ground measurements or LIDAR campaigns should be top priority for governments and intergovernmental organizations responsible for the management of deltaic and near-coastal regions. And evenly important, when high-accuracy elevation datasets are available, they should not be kept confidential within governmental bodies, but made publically accessible, including all metadata, for the scientific community and NGOs. Otherwise researchers may continue to settle for best available, but potentially highly inaccurate, spaceborne DEMs which are in essence unsuitable to use in SLR impact assessments for low-lying, flat deltas. Only when an open-data strategy is adopted, then the next step toward improved SLR impact assessments for low-lying deltas and coastal plains can be made. And these assessments are crucial in directing adaptation and mitigation policies to safeguard world's deltas for future generations.

## Methods

**SRTM digital elevation model.** The SRTM DEM was created from phase-difference measurements of interferometric synthetic aperture radar (InSAR) collected in February 2000 and was the first near-global topography product for the Earth acquired in a consistent way[13]. The SRTM was designed to create a DEM with an absolute vertical accuracy of 16 m and this goal was met as 90% of the absolute elevation errors is <9 m[49]. The SRTM DEM is a digital surface model, describing the elevation of the Earth's surface including objects at the surface, like buildings and vegetation. Therefore, the SRTM DEM has a tendency to over-estimate actual ground surface elevation. There are different versions of the SRTM DEM available and efforts have been made to optimize the SRTM DEM, for example through vegetation removal[50,51]. Previously published SLR impact assessments for the Mekong delta used basic versions of the SRTM DEM, readily available through online portals, and no post-processing steps were performed to optimize the SRTM DEM for the studied area[28,29,38]. As this paper aims to assess the effect of using a basic version of the SRTM DEM for SLR assessments, we also selected a readily downloadable and widely-used version of the SRTM DEM without performing post-processing corrections. We used the SRTM Plus (or void-filled) DEM version 3.0 with an one-arc second grid, approximately ~30 × 30 m and a vertical resolution of 1 m (DEM available through: lpdaac.usgs.gov/data_access/data_pool).

**MERIT digital elevation model.** The high-accuracy global MERIT DEM[37] was developed by removing major error components, i.e. absolute bias, stripe noise, speckle noise, and tree height bias from existing DEMs. For the Mekong delta region, the MERIT DEM uses the SRTM DEM as baseline, and unobserved areas were filled with the Viewfinder Panoramas DEM. The authors report an improvement of vertical accuracy compared to the original DEMs (from 39 to 58% of the land areas mapped with an accuracy of ±2 m or better)[37]. To portray the improvements of the MERIT DEM, and especially the improved landscape representation for flat regions, the Mekong delta was used as example on the data portal website (http://hydro.iis.u-tokyo.ac.jp/~yamadai/MERIT_DEM/). We acquired the MERIT DEM for the Mekong delta from the data portal.

Both the SRTM and the MERIT DEMs contain an obvious elevation error in the southwest corner of the delta where the elevation is <−1 m below MSL (Supplementary Fig. 12 and Supplementary Discussion 1). This part of the SRTM and MERIT DEM was therefore omitted in the DEM quality assessments in this study.

**Topographical elevation points.** We acquired a dataset with almost 20,000 elevation points located in the Vietnamese Mekong delta from the Division of Water Resources Planning and Investigation of the South of Vietnam derived from the national topographical map of 2014 (scale 1:200,000) made by the Department of Survey and Mapping of Vietnam, part of Vietnam's Ministry of Natural Resources and Environment (Supplementary Fig. 1). The dataset has an average point density of 0.51 points per km$^2$. The precision of the elevation indicated on the maps is 0.1 m for elevation points up to +2 m. Between +2 m and +3 m the elevations are given at 0.5 m intervals, elevations higher than +3 m are documented with intervals of 1 m. The data is projected in the VN-2000 coordinate reference system and vertically referenced to the Vietnam's geodetic Hon Dau datum, which has its elevation origin at mean sea level (MSL) of the tide gauge at Hon Dau, an island offshore of Hai Phong in North Vietnam. The applied measurement technique of the elevation points present in the topographical map is not documented but likely derived from geodetic survey and photogrammetric data, as is common practice in Vietnam. A potential offset between mean sea level at the Hon Dau tide gauge (defining its zero datum level) and mean sea level in the Mekong delta cannot be excluded and may introduce additional uncertainty in elevation relative to local sea level along the Mekong. Still, this topographical dataset is presently the only and best available elevation data for the Mekong delta, apart from the global DEMs.

**Interpolation of the topographical elevation model.** We interpolated the elevation points to create a smooth, delta-wide, topographical (Topo) DEM. Based on the optimal points per grid cell[52] and the point density of the dataset (0.51 points km$^{-2}$), a grid cell size of 500 × 500 m is appropriate. A larger cell size would result in an increased RMSE, while a smaller cell size would result in an unfounded higher resolution. We tested several interpolation methods available in the 3D analyst and geostatistical analyst toolbox of ArcMap v.10.3.1., i.e. Inverse distance weighting, ordinary kriging, empirical Bayesian kriging, nearest neighbor, spline and ANUDEM. We compared the resulting DEMs based on a statistical analysis using 120 randomly distributed control points (a subset from the elevation points excluded from the interpolation, see SI, Supplementary Methods 1, Supplementary Fig. 13) and inspection of interpolation correctness in areas with large elevation differences (e.g. spline interpolation created erroneous negative elevations around higher elevated bed rock outcrops). The DEM interpolated using empirical Bayesian kriging employing empirical data transformation and an exponential model produced the lowest mean absolute deviation of all control points (0.22 m) and interpolated realistically between points with larger elevation differences (Supplementary Table 6, Supplementary Fig. 13). Consequently, this method was selected

to interpolate the topographical elevation points and create the Topo DEM (see Supplementary Fig. 14 for interpolation settings).

Before interpolation, all elevation points exceeding individual elevations of +10 m were removed from the dataset. These points are located on highly elevated limestone outcrops towering above the otherwise flat delta plain and including them in the interpolation would inevitably introduce errors to the elevation of the flat delta plain in the immediate surroundings. After interpolation, these areas with elevated limestone bedrock outcrops were excluded from further analyses using a shapefile delineating them based on Google Earth imagery. Furthermore, the large Mekong river branches were removed from the Topo DEM, which is also the case for the SRTM DEM. The Topo DEM resulting from the interpolation has an average root mean square error (RMSE) of 0.16 m that spatially increases with decreasing elevation point density and increasing local elevation variation (Supplementary Fig. 15).

**Validation of absolute elevation**. We evaluated the absolute elevation of the SRTM, MERIT, and Topo DEMs relative to Vietnam's geodetic datum using an independent dataset of 69 national benchmark elevation measurements throughout the delta managed by the Department of Survey and Mapping of Vietnam (see SI, Supplementary Table 2 and Supplementary Fig. 4). The dataset provides coordinates (VN-2000) and vertical elevation at mm precision referenced to Vietnam 2000 geodetic Hon Dau datum (origin at MSL at Hon Dau tide gauge)[53]. The geodetic network of national benchmarks in South Vietnam was built by radio-positioning and traverse measurement techniques connected to stable points with known elevation at bedrock outcrops at the edge of the delta plain[53]. Vertical elevation accuracy of the measurements is not documented. Benchmarks are reportedly located ~30 cm below terrain surface for protection (Supplementary Fig. 5), however it is uncertain whether this is the case for all benchmarks in the dataset. As we compare the elevation of point locations to the average elevation of an entire grid cell (30 m × 30 m for the STRM DEM, 94 m × 94 m for the MERIT DEM and 500 m × 500 m for the Topo DEM), we do expect differences between individual elevation points and the elevation models cells. If the overall elevation of the DEM is in agreement with the overall benchmark elevation, the residuals are expected to show a narrow, non-skewed normal distribution centered at zero.

**Validation of relative elevation**. We assessed the correctness of the spatial distribution of the relative elevation of the DEMs by using two datasets that function as proxy for relative elevation: (i) a geomorphological map[39] and (ii) a flood occurrence map[15]. The geomorphological map of the Vietnamese Mekong delta[39] was mapped using aerial photographs and satellite images combined with field surveys, cored sediment samples and paleoenvironmental assessment using microfossils (Supplementary Fig. 7). It shows the presence and distribution of different geomorphological regions and features throughout the delta. In a natural setting, each geomorphological unit is characterized by a certain elevation relative to other geomorphological units because on differences in depositional environment (Supplementary Fig. 6). For example, natural levees and beach ridges are higher elevated than adjacent flood basins and coastal plains. Therefore, the geomorphological map can serve as proxy for spatial relative elevation distribution. A correct DEM should provide a similar logical elevation pattern, correctly reflecting the relative elevation of the different geomorphological units. We digitized the geomorphological map into a polygon shapefile in ArcMap and extracted the DEMs statistics per geomorphological unit.

We grouped the geomorphological units in three categories that characterize depositional environments with a typical elevation distribution. The first category consists of the Pleistocene geomorphological units, which are mainly found in the N and NW of the Vietnamese Mekong delta. We expect all Pleistocene deposits to be higher elevated than the younger Holocene deposits, because, otherwise, they would have been buried by Holocene onlap deposits. Within the Holocene deposits, we distinguish between the higher elevated, alluvial landscape in the Upper delta plain and the lower elevated coastal plain in the Lower delta plain[54] (Supplementary Fig. 7).

Within each category, we estimated the expected elevation of a geomorphological unit relative to the other units based on typical landscape geomorphology (Table 2; Supplementary Fig. 6A). For the alluvial landscape of the Upper delta plain, we based our expected relative elevation on typical channel belt —floodplain morphology in lowlands[55]. Natural levees are the highest elevated units, followed by channel bars and abandoned channel belts, which are in turn elevated higher than flood basins (partly as a result of post-depositional subsidence of soft flood basin soils). Swamps—requiring frequent flooded and waterlogged conditions—are located in the lowest parts of the landscape. For the coastal environments, we based the expected relative elevation on typical coastal morphology, with mangroves and relict sandy beach ridges separating the tidal flats from the back barrier salt marshes and coastal plain with fresh water marshes in the hinterland (Table 2; Supplementary Fig. 6B). The elevation of tidal flats at the coastline are expected to match high tide levels. Sand spits and especially relict beach ridges are generally elevated higher than the tidal flats. The near-coastal mangrove and salt marshes are expected to have a similar elevation as the tidal flats as they trap sediments during high tide. The back barrier coastal plain is expected to have a lower elevation, as a result of the combination of ongoing compaction of the Holocene strata[25] and reduced sediment supply with progradation of the coastline. Inland marshes within the coastal plain are expected to have the lowest relative elevation, as they are located furthest away from the coastline and active tidal channels, which reduces sediment delivery even further. At delta scale, the coastal plain in the western part of the delta is expected to be lower elevated than the coastal plain in the east, as the nearby Gulf of Thailand only has a tidal range of 40 cm[40] and no direct sediment delivery by rivers.

The second method to validate the spatial distribution of the relative elevation is based on the assumption that lower areas in the flat coastal zone are more often inundated than higher areas, either naturally-induced by flooding or human-induced for agri- and aquaculture purposes Kuenzer et al.[15] created 128 maps of the extent of floods in the Mekong delta from 2007 to 2011 based on Envisat ASAR-WSM (Advanced Synthetic Aperture Radar Wide Swath Mode) satellite images. Combing these maps resulted in a cumulative inundation map showing the number in which an area was inundated during this four year period with a grid cell size of 150 m (Supplementary Fig. 9). The authors distinguished four influencing factors for inundation in the Mekong delta: (1) natural floods of the Mekong river and overland flow, (2) artificial floodwater distributed by canals and controlled by dykes and sluice gates, (3) extreme local precipitation events, and (4) floods related to high tides[15,56]. Inundations resulting from the first three factors do not solely correlate to lowly elevated areas, as they also occur in elevated areas. For example, river floodwater and overland flow happens mostly in the higher, upstream areas as dikes and canals block and retain the floodwater, preventing it to reach the lowest, more distal parts of the delta plain. Extreme precipitation events can occur anywhere in the delta, as the flat topography of the region does not cause orographic lift which would induce increased precipitation at a certain location. However, inundations following tidal flooding are expected to negatively correlate to elevation in terms of extent and duration, as tidal floodwater inundates the lowest topographical areas first and longest. In general, inundation in the northern and middle part of the Mekong delta is predominantly caused by river-induced floods, overland flow and human floodwater control and retention, whereas in the southwestern part of the delta, flooding is induced by both high tides and human action (Supplementary Fig. 9).

We compared the spatial pattern of tide-dominated flood occurrences (which is related to the elevation relative to sea level) to the relative elevation of the DEMs. We only considered those areas where floods are determined by sea water level and tides, which include the provinces of Ca Mau, Bac Lieu, Soc Trang and the southern part of Kien Giang (Supplementary Fig. 9). Part of the area in the southwestern tip of the Mekong delta experience very long, up to year-round, inundation to accommodate aquaculture, mainly shrimp farms[57]. Although inundation of such aquaculture areas is human controlled, e.g. by opening/closing of sluices[58], they are located in the lowest parts in the landscape, to facilitate water circulation and management. Therefore, the presence of aquaculture does not obstruct the correlation between inundated area and elevation.

The analysis was performed by sampling the elevation of the DEMs at the center of each inundation-map raster cell and calculating the elevation statistics per inundation occurrence. Less than one percent of the total area is inundated more than 25 out of 128 times. As the areas per inundation occurrence >25 became too small to derive a representative mean elevation from the DEMs, they were excluded from the analysis.

**Sea-level rise impact assessments**. To evaluate the effect of using the SRTM, the MERIT or the Topo DEM for relative SLR (hereafter: SLR) impact assessments, we estimated the area below sea level after future SLR for each DEM. Both the SRTM and the MERIT DEM are referenced to WGS84 and EGM96. Zero vertical elevation in these DEMs represents zero elevation to the global earth gravitational model (EGM) and this likely differs from the local tidal datum, and thus sea level. Nonetheless, numerous previous studies directly used the SRTM DEMs elevation for SLR impact assessments, erroneously assuming 0-m elevation (to EGM96 datum) to equal local mean sea level[28,29,38]. To evaluate to what extent such assumption would lead to errors in SLR impact assessments, for sake of discussion, we purposely assumed 0-m elevation in both STRM and MERIT DEMs to represent mean sea level, thereby mimicking previous studies. Although actual MSL in the Mekong delta possibly also differs from the Hon Dau tidal datum, in the absence of additional data, we assumed the zero elevation in Hon Dau datum to represent current MSL in the Mekong delta. For the purpose of analysis of comparing the performance of the Topo and the MERIT DEMs in SLR impact assessments, we attempted to account for the difference in vertical datum by vertically shifting the MERIT DEM to match the Topo DEM's mean elevation of the Mekong Delta. This was done by subtraction of the absolute difference in mean elevation of the two DEMs, in this case 2.5 m.

The vertical resolution of both the MERIT and the Topo DEMs allowed detailed quantification of the area affected by rising sea level. We quantified the areas falling below sea level for SLR scenarios of 20, 50, 80, and 100 cm. In case of the SRTM DEM with a vertical resolution of 1 meter, we determined the area falling below sea level for a SLR of 1 meter, similar to analyses done in previous studies[28,29,38]. We also estimated the number of people living in the area below sea level for each scenario by using provincial population statistics of 2016 (people per km²; Supplementary Table 7). As spatial data on the population distribution within provinces was not available, we assumed an even distribution excluding rivers and steep bedrock outcrops. Additionally, a detailed analysis of delta surface and people impacted for each province individually was done for the Topo DEM.

## Data availability
The Topo DEM of the Mekong delta (Fig. 2) is publicly available as ASCI file in the Pangaea online repository[59]: https://doi.pangaea.de/10.1594/PANGAEA.902136. The topographical elevation points dataset is available upon request.

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

## Acknowledgements

The Division of Water Resources Planning and Investigation of the South of Vietnam (DWRPIS) is thanked for digitizing the dataset of the topographical elevation points used to create the Topo DEM and providing data on the independent elevation benchmarks in the Mekong delta. Ramon Hansen is thanked for discussing geodetic aspects of the analyses. Lau Nhuyen Ngoc from the Ho Chi Minh City University of Technology is thanked for sharing documentation and discussions on Vietnamese coordinate reference systems and geodetic datums. John Shaw and an anonymous reviewer are thanked for constructive comments on the manuscript. This research is part of the Urbanizing Deltas of the World (UDW): "Rise and Fall" research project (grant: W 07.69.105) funded by the Dutch scientific organization (NWO-WOTRO), Deltares and the TNO—Geological Survey of The Netherlands.

## Author contributions

P.S.J.M. acquired the elevation dataset, conceived the study, and designed the analyses. L.C., P.S.J.M. and E.S. performed the analyses. P.S.J.M., L.C., G.E., H.M. and E.S. analyzed and discussed the data. P.S.J.M. drafted the paper, which was then reviewed by all co-authors.

## Additional information

**Competing interests:** The authors declare no competing interests.

