## [Peer Review File · Nature Communications]

Reviewers' comments:

Reviewer #1 (Remarks to the Author):

This paper analyses the effects of two DEMs (SRTM) and a topographic contour lines derived DEM (Topo-DEM) to examine sea level rise impacts. The focus area is the Mekong Delta. The authors claim that the Delta is on average almost two meters lower than previously thought.

The paper is extremely well written and it was a very interesting read. The paper should be of interest to a wide range of readership. The methods used are sound and the conclusions are supported by the analysis.

However, there are two major points of concerns here and they primarily link to the following two papers that are not cited by the authors:

1) Yamazaki et al. (2017) A high-accuracy map of global terrain elevations, <https://agupubs.onlinelibrary.wiley.com/doi/full/10.1002/2017GL072874>

2) Kulp and Strauss (2016) Global DEM Errors Underpredict Coastal Vulnerability to Sea Level Rise and Flooding <https://www.frontiersin.org/articles/10.3389/feart.2016.00036/full>

Yamazaki et al (2017) present a much improved version of SRTM-DEM and shows that especially in Deltas and the Mekong included that version is much improved. The "striping" that one sees on the authors SRTM figures is a SRTM-based effect and can be removed, which Yamazaki et al in their MERIT version have done.

In other words, I do not think this paper should be accepted without the authors using MERIT for instance. Also there are other free DEMs out there like ASTER which the authors could also have assessed, although I do suggest to try MERIT instead. This DEM is freely available for non-commercial purposes (at this point in time). An access password can be requested from Dai Yamazaki.

The second paper I listed, authors Kulp and Strauss also used SRTM (note that their work was published before the improved MERIT version was released) and concluded the same, namely SRTM underpredicts coastal vulnerability because its heights are largely inaccurate in coastal areas.

I think the authors need to use MERIT and see how that compares to the Topo-DEM and then also need to acknowledge the work of Kulp and Strauss and demonstrate how the work by the authors in this paper is different from theirs.

Reviewer #2 (Remarks to the Author):

Review of "Mekong delta much lower than previously thought: accurate elevation data crucial to sea level rise impact assessments in deltas" by Minderhoud et al.

By John Shaw, University of Arkansas (shaw84@uark.edu)

This study describes a new Digital Elevation Model (DEM) built from about 20,000 elevations surveyed by the Department of Surveying and Mapping of Vietnam. The authors argue that this DEM is about 2 m lower than a DEM created from SRTM data, which has coarser vertical resolution and clear artifacts. The authors use this new dem to make many important conclusions about inundation of land and human populations. If true, this paper significantly changes how urgent rising sea levels are for Vietnam.

However, I am not convinced of this conclusion. The new point elevations are presented without any reference or metadata, making the authors guess their vertical datum. This is insufficient for a study of this importance. For me to be convinced, I need to see a leveling comparison of EGM96, which I read to be the vertical datum of the SRTM data that tracks sea level with cm-scale precision, and whatever the surveyed elevations vertical datum.

From my studies of deltas in coastal Louisiana, I know that there are many choices for vertical datum, some of which are tied to sea level, and some of which are not. The variation among them is several decimeters, including for NAVD88, which is the most common surveying datum in North America. Here are the variations among datums at a particular tide gauge in Louisiana (<https://tidesandcurrents.noaa.gov/datums.html?units=1&epoch=0&id=8764227&name=LAWMA%2C+Amerada+Pass&state=LA>). The validation with the benchmarks was an important step, but if they were from the same datum in as the points and both varied from EGM96, then it provides no real validation for this study.

In 10 minutes of searching, I have not found a similar comparison of vertical datums for the Mekong (and really need the vertical datum of the points anyway). I imagine that comparison is possible though with geodetic field work on the delta, or perhaps with a combination of satellite and ground truth data (<https://sealevel.jpl.nasa.gov/missions/topex/>). I cannot support acceptance until a direct comparison between the datums of the old and new DEM can be made.

When discussing inundation, the authors neglect to mention that the Mekong delta has a mean tidal range of 2.6 m (Syvitski, 2005). This means that the entire 1 m of sea level rise analyzed by the authors comprises area that is either flooded daily or sufficiently protected. I do not mean to say that sea level rise poses no danger. However, I suggest that the gradually rising seas be discussed in terms of nuisance flooding, the increasing frequency by which high tides flood important land (Moftakhari et al., 2015, 2017).

The 2.6 m tidal range also influences the interpretation of Kuenzer et al.'s flooding data. Figure 5 shows that the 15 maps with the largest flooding cover an elevation of about 1.6 m EGM96. 1.6 m is just over half of the tidal range. One could interpret these results as high tides covering the elevations that they should cover, and perhaps validating the SRTM data.

Since the paper relies on the interpolated DEM, I also urge the authors to provide the details of the kriging procedure (in the supplementary material). This includes the semivariogram with sill, range, and nugget labeled, and some form of cross-validation, to check for biasing in the interpolated result. These can all be done in ArcMap's Geostatistical toolbox. This information would make the work more reproducible.

These constitute my major concerns of this manuscript. Beyond this, I was generally impressed with the figures, analyses and care mentioning all processes that can change land elevation (page 18). If these points can be addressed, I would support this manuscripts publication in Nature Communications.

Minor Comments

I support publishing all the data associated with this article in some sort of repository (e.g. figshare.com). Given the authors admonition to make all data freely available to the international research community, I recommend the authors do the same. It would certainly boost the import and eventual citations of this manuscript.

Page 1 (RSLR) instead of (RSL)

Page 3: maps from these studies, indicating flood zones, have been used... instead of "were been used"

Table 1: If the new elevation is given in decimeter scale, I think that analyses should provide only decimeter outputs. Same with SRTM: can't tell better than 1 m.

Table 3: I recommend adding a note that there are currently surveyed areas that are below 0 m, which is why the present scenario values (columns 1 and 3) aren't zero.

Page 18: the final sentence "what to protect and what not" sounds quite flippant to me. I would not write it for such dangerous news. I'll leave the choice to the authors though.

Page 22, line 3: (051 points km⁻²) is how I think this should be presented, no?

Page 29, line 2: (SLR).

References

Moftakhari, H. R., AghaKouchak, A., Sanders, B. F., Feldman, D. L., Sweet, W., Matthew, R. A., & Luke, A. (2015). Increased nuisance flooding along the coasts of the United States due to sea level rise: Past and future. *Geophysical Research Letters*, 42(22), 9846–9852.

<https://doi.org/10.1002/2015GL066072>

Moftakhari, H. R., AghaKouchak, A., Sanders, B. F., & Matthew, R. A. (2017). Cumulative hazard: The case of nuisance flooding. *Earth and Space Science*, 214–223.

[https://doi.org/10.1002/2016EF000494@10.1002/\(ISSN\)2333-5084.SCISOC1](https://doi.org/10.1002/2016EF000494@10.1002/(ISSN)2333-5084.SCISOC1)

Syvitski, J. P. M. (2005). The morphodynamics of deltas and their distributary channels. In G.

Parker & M. H. García (Eds.), *River, Coastal Plain and Estuarine Morphodynamics: RCEM 2005z* (Vol. 1, pp. 143–150). London: Taylor & Francis Group. Retrieved from

<http://books.google.com/books?hl=en&lr=&id=AXwMpddTRAAC&oi=fnd&pg=PA143&dq=The+morphodynamics+of+deltas+and+their+distributary+channels&ots=L3KCGb9aSu&sig=MCwbjukOJ6VIE2pM4RLunTqdnco>

General response to reviewers remarks

We are very grateful for the input of both reviewers as their constructive comments helped us to significantly improve the manuscript and strengthening the main message. Two major advancement have been made in this revised version: 1) the global MERIT DEM was included in all analyses (main point, reviewer 1) and the vertical datum of the Vietnamese elevation points was uncovered and included throughout the entire manuscript (main point, reviewer 2).

By adding the newer (SRTM-derived) MERIT DEM to the analysis, a comparison to a state-of-the-art, error-removed global DEM could be drawn with the new Topo DEM. We found that visually the MERIT DEM creates a more appealing and smooth image of the Mekong delta compared to the SRTM and it does get rid of some of the obvious elevation errors. At the same time, the MERIT DEM is based on SRTM data and thus still limited by some of the fundamental errors of the SRTM DEM.

Furthermore, our continued search to find metadata on the Vietnamese vertical datum of the elevation points was eventually successful and it made us become aware of a new, more fundamental issue related to the use of global DEMs. This is the potentially large mismatch between global vertical datum of these DEMs and local tidal datum. We feel that this insight increases the impact and timeliness of this revised manuscript even further. Not only provides it a fundamental explanation to why the Mekong delta was, based on the SRTM DEM, previously (erroneously) assumed to be much higher elevation than it actually is, this revelation also spells potential huge errors on the estimated elevation of deltas and coastal regions elsewhere in the world based on global elevation models. This subsequently affects all sea-level rise and coastal (e.g. storm surge) flooding studies that used global DEMs and omitted the step to convert this data to local tidal datum (and this is often not done, either because the necessary local data is lacking or it is neglected out of ignorance or because it is not perceived to be significantly influencing results). Our study suggests potential errors up to several meters in data-sparse coastal regions around the world.

The main conclusions that we drew on the elevation of the Mekong delta as presented in the initial submission still stand and we feel that they are even strengthened further after addressing all remarks raised by the reviewers. Please find our respond to each individual point raised by the reviewers below.

Point-by-point response to the comments and suggested changes by referee 1

Reviewer #1 is thanked for the positive review and we are very pleased with the following comment: *'The paper is extremely well written and it was a very interesting read. The paper should be of interest to a wide range of readership'*. Reviewer #1 brings up two major points of concern: 1) the MERIT DEM, which is an improved version of the SRTM DEM by Yamazaki et al. (2017) and 2) the paper by Kulp and Strauss (2016):

Reviewer #1, point 1:

The reviewer suggests to include the MERIT DEM into the analysis and we fully agree with this suggestion as the MERIT DEM claims to fix many errors present in the SRTM DEM. In the updated manuscript, we included the MERIT DEM in all analyses and discussions and we feel that this greatly improved the manuscript.

For the reviewers convenience, here we shortly summarize the findings and conclusions related to adding the MERIT DEM in our analyses:

We found that visually the MERIT DEM creates a more appealing and smooth image of the Mekong delta, while at the same time, MERIT is still fundamentally based on SRTM data and is thus limited by the errors in SRTM. This can be seen by the peculiar elevation patterns that are still present in the MERIT DEM, although smoothed (Figure 3). In our validation of the relative elevation using the geomorphological map, the MERIT DEM performs worse than the original SRTM DEM, while for the inundation patterns, the MERIT DEM performs better. When the DEM is used in terms of inundation assessments (we also used a transposed version of the MERIT DEM to correct for difference in vertical datum), the results are actually not much better than when using the SRTM DEM, although visually they look more appealing as the MERIT DEM successfully got rid of a striping pattern present in the SRTM DEM (Figure 8). The conclusions drawn in the first version of the manuscript on the comparison of the Topo DEM with the SRTM DEM are also valid for the comparison with the MERIT DEM.

The reviewer also suggests to use for example other DEMs, such as the ASTER. We did assess the suitability of the ASTER DEM in an early phase of this study, but as the results of the ASTER DEM were so mediocre compared to the SRTM DEM (the reason why SRTM is widely used and ASTER is not), we decided to exclude the ASTER DEM completely and focus on the SRTM DEM, and in this revision also its derivative, the MERIT DEM.

Reviewer #1, point 2:

Reviewer #1 suggests the work by Kulp and Strauss (2016) to be acknowledged in the manuscript and to show the difference with our work. We gladly did so. Kulp and Strauss compared SRTM-based DEMs with high-accuracy LiDAR data in the United States, which is a very data rich region. Most regions in the world, like the Mekong delta, are located in data sparse regions. What furthermore sets our work apart is that we reveal the mis-use of global DEMs when ignorance of lack of data causes researchers to omit conversion of the global DEMs vertical datum to a local geodetic and tidal datum prior to the analyses

(see also response to Reviewer #2). We included the following references in the manuscript to Kulp & Strauss (2016):

Line 36-37:

'Few studies specifically focused on the effect of DEM accuracy on assessments of impacts of sea-level rise on low-lying flat coastal areas¹² (Kulp and Strauss, 2016).'

Line 345-347:

'These results confirm the observation that, even when correctly converted to local tidal datum, SRTM-based estimates have the tendency to underestimate sea-level rise impacts for lower elevated coastal areas due to DEM errors (Kulp & Strauss (2016).'

Line 415-416:

'Presently available spaceborne DEMs suffer from accuracy problems in the order of meters, and will lead to major errors in assessments of coastal drowning (Kulp & Strauss, 2016).'

Furthermore, we also included other new, relevant literature on this subject, published in December 2018, in the revised manuscript:

Hawker, L., Bates, P., Neal, J., Rougier, J., 2018. Perspectives on Digital Elevation Model (DEM) Simulation for Flood Modeling in the Absence of a High-Accuracy Open Access Global DEM. Front. Earth Sci. 6, 233. doi:10.3389/feart.2018.00233

Schumann, G.J.-P., Bates, P.D., 2018. The Need for a High-Accuracy, Open-Access Global DEM. Front. Earth Sci. 6, 225. doi:10.3389/FEART.2018.00225

Point-by-point response to the comments and suggested changes by referee 2

Reviewer #2 (John Shaw) is thanked for the extensive review and kindly pointing us in the right direction in thinking about vertical datums. We are glad with the following statement *'If these points can be addressed, I would support this manuscripts publication in Nature Communications.'* The reviewer raised three main points of which the most important point is the lack of information on vertical datum of the Vietnamese elevation points. We comment on these points below:

Reviewer #2, point 1:

The new point elevations are presented without any reference or metadata, making the authors guess their vertical datum.

We agree that this point was raised, as this also bothered us (and which made us go through such extensive validation analyses in the first place). Fortunately, after continuing our search, we finally managed to get the required information on the Vietnamese vertical datum used for the elevation points and their relations to the EGM96 and EGM08 models. Mainly thanks to discussions and Vietnamese literature supplied by Dr. Lau Nhuyen Ngoc from the Ho Chi Minh City University of Technology, his efforts are mentioned in the acknowledgement of the revised manuscript). The elevation points, used to produce the new Topo DEM, and the independent benchmarks, used in absolute elevation validation, are both projected in the VN-2000 coordinate reference system and vertically referenced to the Vietnam's geodetic Hon Dau datum, which has its elevation origin (zero) at mean sea level (MSL) of the tide gauge at Hon Dau, an island offshore of Hai Phong in North Vietnam. This means that the Hon Dau datum is a tidal datum, which make the elevations referenced to MSL.

The new metadata and literature search on this topic made us aware of the fundamental issue related to global EGM and local vertical datum. In order to used global DEMs to get information of elevation to local sea level, the vertical datum has to be converted to local tidal datum. This step is often neglected, partly because of the wide-spread assumption that the EGM96 already tracks sea level up to dm precision (Reviewer 2 also stated: *EGM96, which I read to be the vertical datum of the SRTM data that tracks sea level with cm-scale precision*). We found that this wide-spread assumption may be true for the United States, but is incorrect for other parts in the world. A general statement on the EGM is the following: *'The Earth Geodetic Model (EGM96), developed in a collaborative effort by NASA Goddard Space Flight Center, the National Imagery and Mapping Agency (NIMA), and Ohio State University, has been used to compute geoid undulations accurate to better than one meter (except in areas lacking accurate surface gravity data). The values of this surface show at every location how much MSL varies from the ellipsoid.'* (source: <https://www.esri.com/news/arcuser/0703/geoid2of3.html>). Key here is the statement between brackets: **except in areas lacking accurate surface gravity data**. This may actually be the case for a lot of data-sparse regions in the world.

We added the following to the revised manuscript 56-67: *'A second issue arises from the fact that when global DEMs are used for assessments of relative elevation to local sea level, the elevations need to be converted from their reference to a global geoid to a local datum referenced to sea-level height²⁰. However, in many global and regional studies, this crucial step is very often neglected, either due to lack of data on local tidal datums or as a result of lack of geodetic expertise. The geoid to which global DEMs are referenced is often assumed to represent global mean sea level, while this is actually not the case for many places in the world. For example in Turkey, vertical offsets up to five meters have been documented between the EGM96 (to which the SRTM DEM is referenced) and the newer EGM06 geoid model²², illustrating the potential magnitude of vertical error. Omitting conversion of DEM elevation to local tidal datum, and thereby, erroneously, assuming geoid datum of global DEMs to represent local sea level introduces potentially large errors in elevation to sea level estimates, especially in areas with strong regional deviations in global earth gravitational models (EGMs).'*

Vietnam is located on a steep slope of the earth gravitational field and in the Vietnamese Mekong delta alone the EGM has already a difference of 10 meters from East to West. A difference of some meters between the global EGM and the actual local gravitational field/sea level is therefore not unthinkable. In Vietnamese geodetic literature we found evidence that this is indeed the case, even for the newer EGM08. We added this statement to the revised manuscript (114-116): *'A comparison in Vietnam between the newer EGM08 geoid and Vietnam's Hon Dau datum revealed a mean elevation bias of +0.890 m, which suggests potentially, similar or even vertical large errors for the EGM96 (Hoa, 2017).'*

This large difference between vertical reference datums is very likely the explanation of the vertical offset that we found between the average SRTM and MERIT DEM elevations (EGM96 datum) and the Topo DEMs elevations (Hon Dau tidal datum). This finding makes the impact of our study even larger as many researchers worldwide who apply elevation data from global DEMs as input data for their field of expertise, are not enough aware of this issue or do not have the required geodetic expertise. They follow the general assumption that the EGM96 of the SRTM DEM provides a good enough approximation of local sea level. With the case of the Mekong delta we show that this may be far from the case. This adds an additional, potentially large, error to elevation estimates of coastal plain and deltas in data-sparse regions worldwide, and affects all impact assessments that are made using global DEM data that is not converted to local tidal datum.

In the revised version, we also analyze the effect of a SLR impact assessment using a global DEM that is corrected for the vertical offset to local tidal datum. We use a transposed version of the MERIT DEM, in which we mimic the effect of a vertical datum conversion to Hon Dau tidal datum by subtracting the average elevation difference between the MERIT and the Topo DEM for the Mekong delta. The results of this analysis can be seen in Fig. 6, Fig. 8 and Table 3.

Reviewer #2, point 2a:

When discussing inundation, the authors neglect to mention that the Mekong delta has a mean tidal range of 2.6 m (Syvitski, 2005). This means that the entire 1 m of sea level rise analyzed by the authors comprises area that is either flooded daily or sufficiently protected. I do not mean to say that sea level rise poses no danger. However, I suggest that the gradually rising seas be discussed in terms of nuisance flooding, the increasing frequency by which high tides flood important land (Moftakhari et al., 2015, 2017).

While the reviewer is correct that the tidal range is around 2.6 m (meaning a tidal amplitude of 1.3 m) in the South China Sea (east of the Mekong delta). This value does not reflect the tidal range for the western part of the delta in the Gulf of Thailand, where the tidal range is as low as 40 cm. (see figure below from Tomkratoke et al., 2015).

Fig. 6. Spatial distribution of co-tidal line (amplitude) and overlaid co-range

This corresponds also to the elevations of the delta shown by the Topo DEM and in the analysis of relative elevation of geomorphological features in the coastal landscape (Table 2, Fig. 10 & 11). Towards to east coast the delta has higher elevations at the coastline as a result of the larger tidal range. The low tidal range in the Gulf of Thailand is also reflected by the absence of higher morphological features and elevations. We agree with the comment of the reviewer on nuisance flooding and added a comment on this in line 424 and included a reference to Moftakhari et al., 2017.

Reviewer #2, point 2b:

The 2.6 m tidal range also influences the interpretation of Kuenzer et al. 's flooding data. Figure 5 shows that the 15 maps with the largest flooding cover an elevation of about 1.6 m EGM96. 1.6 m is just over half of the tidal range. One could interpret these results as high tides covering the elevations that they should cover, and perhaps validating the SRTM data.

See previous point, with all respect, this reasoning is incorrect. The 2.6 m tidal range is only true for the East China Sea and as the eastern coastline exists of higher elevated tidal flats and coastal beach ridges / sand dunes, the lower hinterland on which the analysis focuses (Fig. 12) is not influence by tidal flooding from the East China Sea. This area is however influenced by the sea level in the Gulf of Thailand as the western coastline doesn't have a protective morphology. But since the tidal range here is only ~40 cm (Line 257 and 573), we stand by our interpretation of the results presented in Figure 5.

Reviewer #2, point 3:

Since the paper relies on the interpolated DEM, I also urge the authors to provide the details of the kriging procedure (in the supplementary material). This includes the semivariogram with sill, range, and nugget labeled, and some form of cross-validation, to check for biasing in the interpolated result. These can all be done in ArcMap 's Geostatistical toolbox. This information would make the work more reproducible.

We have included information on all the interpolations methods which we tested, validated and compared in order to come to the best method for interpolation in the supplementary (Table S1). We also provide all required information on the kriging settings used in ArcMap to make the interpolation of the elevation points reproducible (Fig. S2). For the reviewers information, the final method which we applied, empirical bayesian kriging, does not rely on a single variogram, but rather calculated an ensemble of variograms for each location. For more information on the method: https://desktop.arcgis.com/en/arcmap/latest/extensions/geostatistical-analyst/what-is-empirical-bayesian-kriging-.htm#ESRI_SECTION1_FD04B0DC8B734D74AB3208BFE06D1AB5

Minor Comments

I support publishing all the data associated with this article in some sort of repository (e.g. figshare.com). Given the authors admonition to make all data freely available to the international research community, I recommend the authors do the same. It would certainly boost the import and eventual citations of this manuscript.

We have agreed with our Vietnamese partners who provided us the data that we can share the derived DEM product using an online repository. The dataset of elevation points are available upon reasonable request.

Table 1: If the new elevation is given in decimeter scale, I think that analyses should provide only decimeter outputs. Same with SRTM: can 't tell better than 1 m.

As the results are mean values calculated over larger areas, we feel that by reporting these values with an extra digit provides additional insight in the average elevation claimed by a DEM for that area. E.g. 0.6 m and 1.4 m average elevation for two different areas according to a global DEM would otherwise result in 1 m and 1 m, crippling the ability to spot the difference. We added the follow statement to alert the reader to this in the captions of both table 1 and 2: *“The mean values are given with one additional decimal than present in the input data to distinguish mean elevation differences.”*

We included all further textual edits and additions suggested by reviewer 2.

References:

Tomkratoke, S., Sirisup, S., Udomchoke, V., Kanasut, J., 2015. Influence of resonance on tide and storm surge in the Gulf of Thailand. Cont. Shelf Res. 109, 112–126. doi:10.1016/j.csr.2015.09.006

Reviewers' comments:

Reviewer #1 (Remarks to the Author):

In my opinion, the authors have addressed my previously raised concerns to a satisfactory level and to me this paper is now acceptable for publication.

Especially, the points about the MERIT DEM, vertical local datum and the Kulp et al. paper comparison is now included, which I welcome.

Reviewer #2 (Remarks to the Author):

Review of "Mekong delta much lower than previously assumed in sea-level rise impact assessments"

By John Shaw (shaw84@uark.edu)

This is my second review of this manuscript. I find that the authors have sufficiently addressed my previous comments. I have one (somewhat major) stylistic comment, and several minor comments, but I think that the paper is largely good to go. The impact is certainly very large if important decisions are being made with SRTM data that assumes the EGM96 datum is mean sea level in Vietnam.

The major comment is that much of the new analysis feels like straw-man arguments (arguing against a position that is clearly false). Once it has been shown that EGM96 datum departs from the Hon Dau tidal datum, then each analysis that shows the SRTM and Merit DEMs departing from the TOPO DEM (distributions of inundated province area, depth with inundation of one meter) are somewhat elementary. Text like L195-197 is the best example: the SRTM and MERIT data definitely are higher relative to their datum, so it is not surprising that they don't compare well to Hon Dau benchmarks. Also, line 317 and 359 needs to acknowledge the datum mismatch. I do think that it is valuable to show the consequences of assuming EGM96 is sea level, but each of these analyses are written like they independently refute the validity of the satellite based DEMs. I recommend minor revisions to address this.

Minor comments:

My reading into the Hon Dau vertical datum is that it was constructed relative to Mean Sea Level at Hon Dau in 1990. I think it is reasonable to include eustatic sea level rise over the 24 years between 1990 and 2014, when the TOPO DEM points were collected. With 3-4 mm/yr eustatic sea level rise (or an appropriate number for the South China Sea), this would mean that current mean sea level that is 7.2 cm higher than it was in 1990. Perhaps this is too complicated to add to your analysis which only claims to reach decimeter scale (very reasonably), but it would make inundation statistics slightly worse.

Line 14: How well _the_ vertical datum matches sea level

Line 26: Loose elevation _relative_ to sea level

L67: Photogrammetric data represented on topo maps? I am still uncomfortable that the method of measurement for these data remains unknown.

Line 78: I don't understand "a.o."

Line 112: Although I know that converting to Hon Dau is difficult and you have done your best, it is quite easy to convert between EGM96 and EGM08. Here is a handy applet:

<https://geographiclib.sourceforge.io/cgi-bin/GeoidEval>. It looks as if EGM96 is about 22 cm higher than EGM08 around the Mekong delta.

L195 By nature geomorphological units... awkward phrasing

Table 3: Should be 10^6 , not -6

L785: no date is given for Dang et al.

Response to reviewers' comments

We are pleased to hear that both reviewers found that the revised manuscript addressed their earlier concerns well and that both reviewers now support publication.

Reviewer #1 is satisfied with the revised manuscript and has no further suggestions.

Reviewer #2 says that *"the authors have sufficiently addressed my previous comments ... the paper is largely good to go"*. The reviewer has one larger stylistic comment and several minor comments, which we address below.

Stylistic comment: "Much of the new analysis feels like straw-man arguments (arguing against a position that is clearly false). Once it has been shown that EGM96 datum departs from the Hon Dau tidal datum, then each analysis that shows the SRTM and Merit DEMs departing from the TOPO DEM (distributions of inundated province area, depth with inundation of one meter) are somewhat elementary. Text like L195-197 is the best example: the SRTM and MERIT data definitely are higher relative to their datum, so it is not surprising that they don't compare well to Hon Dau benchmarks. Also, line 317 and 359 needs to acknowledge the datum mismatch. I do think that it is valuable to show the consequences of assuming EGM96 is sea level, but each of these analyses are written like they independently refute the validity of the satellite based DEMs. I recommend minor revisions to address this."

We agree with the reviewer that some result-describing sections do not clearly acknowledge the (potential) influence of datum mismatch as underlying principle as possible reason for vertical elevation mismatch, next to overall DEM inaccuracy. We follow the reviewer's suggestion, making the following revisions to acknowledge the datum mismatch (*green is added text*):

*L175-177: "The large mean deviations indicate a structural overestimation of the geodetic surface elevation in the SRTM and MERIT DEMs, *which is partly the result of the difference between Hon Dau datum and the EGM96 to which both DEMs are referenced.*"*

*L186-188: "Whereas the SRTM and the MERIT DEM seem to systematically overestimate actual delta surface elevation, *partly as a result of a different vertical datum*, the Topo DEM appears to represent the geodetic elevation of the delta surface at decimeter accuracy."*

*L258-260: "The MERIT and SRTM DEM, *referenced to EGM96 rather than a tidal datum*, provide absolute elevation values that are at least one meter higher ..."*

L314-320: In this section the order of argumentation is changed: first stating that the vertical reference datum is different, before discussing the overestimation of absolute elevation:

Original: "Whereas the SRTM and MERIT DEM overestimate the absolute surface elevation at the national benchmarks on average by 2 and 3 m, the Topo DEM shows almost no absolute elevation overestimation (0.2 m). A fundamental explanation for this can be found in the difference in vertical datum. The Topo DEM and national benchmarks are referenced to Hon Dau datum, while both the SRTM and the MERIT DEM are referenced to the global EGM96 geoid."

Revised: *“The SRTM and the MERIT DEM are both vertically referenced to the global EGM96 geoid while the Topo DEM and the national benchmarks, used to evaluate absolute vertical elevation, are referenced to Hon Dau datum. Partly for this reason, the SRTM and MERIT DEM overestimate the absolute surface elevation at the national benchmarks on average respectively 2 and 3 m, while the Topo DEM shows almost no absolute elevation overestimation (0.2 m).”*

L352-354: *“According to the Topo DEM, the Mekong delta plain has a mean elevation of 0.82 m above Hon Dau tidal datum, which is only one third of the SRTM DEM’s mean elevation of 2.6 m, and even less compared to the MERIT DEM’s mean elevation of 3.3 m. Both SRTM and MERIT DEMs are referenced to EGM96 datum, of which zero elevation was in previous studies assumed to match sea level, while actual local sea level seems to be much higher.”*

For the sea-level rise assessments, the assumed relation between vertical datum and sea level is clearly stated in the method section: *“To evaluate to what extent such assumption [i.e. erroneously assuming zero meter elevation (to EGM96 datum) to equal local mean sea level] would lead to errors in SLR impact assessments, for sake of discussion, we purposely assumed zero meter elevation in both STRM and MERIT DEMs to represent mean sea level, thereby mimicking previous studies.”*

Minor comments Reviewer #2:

My reading into the Hon Dau vertical datum is that it was constructed relative to Mean Sea Level at Hon Dau in 1990. I think it is reasonable to include eustatic sea level rise over the 24 years between 1990 and 2014, when the TOPO DEM points were collected. With 3-4 mm/yr eustatic sea level rise (or an appropriate number for the South China Sea), this would mean that current mean sea level that is 7.2 cm higher than it was in 1990. Perhaps this is too complicated to add to your analysis which only claims to reach decimeter scale (very reasonably), but it would make inundation statistics slightly worse.

The reviewer is right and we are aware of this aspect. However, it is also unclear whether the zero level of the Hon Dau datum has been updated since 1990 (probably not, but adjustment cannot be excluded). As the reviewer already points out, the effect of past eustatic sea level rise since Hon Dau is smaller than the accuracy of the analysis and the elevation model (decimeter), which was the reason not include this aspect. In our sea-level rise assessments, the possible effect of *past* eustatic sea level rise since 1990 is already tackled by the following assumption: L633-635: *“Although actual MSL in the Mekong delta possibly also differs from the Hon Dau tidal datum, in the absence of additional data, we assumed the zero elevation in Hon Dau datum to represent current MSL in the Mekong delta.”*. We added *“current”* to the assumption to make this clearer.

L67: *Photogrammetric data represented on topo maps? I am still uncomfortable that the method of measurement for these data remains unknown.*

We could not agree more with this statement by the reviewer that this is not ideal, but this is the information that was consistently given on the data points by different Vietnamese government institutes. Before LIDAR data became available, producing elevation points on topographical maps using photogrammetric was the standard practice for decades around the world. Therefore, we believe this is indeed the source of the topographical data points in Vietnam.

Line 88: I don't understand "a.o."

To avoid confusion, we have changed a.o. into 'amongst others' throughout the text.

Line 112: Although I know that converting to Hon Dau is difficult and you have done your best, it is quite easy to convert between EGM96 and EGM08. Here is a handy applet:

<https://geographiclib.sourceforge.io/cgi-bin/GeoidEval>. It looks as if EGM96 is about 22 cm higher than EGM08 around the Mekong delta.

The sentence in the manuscript referred by the reviewer (L110-112. "A comparison in Vietnam between the newer EGM08 geoid and Vietnam's Hon Dau datum revealed a mean elevation bias of $+0.890\text{ m}^2$, which suggests potentially, similar or even vertical larger errors for the EGM96.") points out that also in Vietnam the difference between the two global EGM's and local geodetic datum can be rather large (similar to other places on earth).

We thank reviewer 2 to point us to this application tool that allows converting between different EGM datums for a single coordinate. Also there is ample literature on EGMs and comparisons between them. If one would want to investigate the difference between different EGMs for an area, an areal comparison should be done rather than for a single coordinate, as EGM models can spatially change considerably. In the context of our study, quantifying the difference between EGM96 and EGM08 over the Mekong delta is not relevant, as the main point is that both divert from the local geodetic datum.

L785: no date is given for Dang et al.

We have added the publication date (2010).

Further minor textual comments:

We have accepted all suggestions for textual changes.